# Work-life balance in physicians working in two emergency departments of a university hospital: Results of a qualitative focus group study

**Samipa Pudasaini**[1], **Liane Schenk**[2], **Martin Möckel**[1], **Anna Schneider**[2]*

**1** Department of Emergency Medicine, Campus Virchow-Klinikum and Campus Charité Mitte, Charité–Universitätsmedizin Berlin, Corporate Member of Freie Universität Berlin and Humboldt-Universität zu Berlin, Berlin, Germany, **2** Institute of Medical Sociology and Rehabilitation Science, Charité–Universitätsmedizin Berlin, Corporate Member of Freie Universität Berlin and Humboldt-Universität zu Berlin, Berlin, Germany

* anna.schneider@charite.de

**Data Availability Statement:** Data cannot be shared publicly because of legal and ethical restrictions. Our focus groups contained person-related sensitive information and the

## Abstract

By applying an explorative approach, we aimed to identify a wide set of challenges and opportunities for the compatibility of the work and life domains in emergency department (ED) physicians as well as their suggestions for practical approaches to improve work-life balance. Four focus groups with 14 physicians of differing hierarchical position and family status were carried out at two EDs of one major university hospital. Data analysis was based on qualitative content analysis. Discussed themes within main categories included ED work conditions, aspects of residency training, physician's mentality and behaviors as well as context factors of university medicine. Working in an ED is associated with a comparatively high work-life-interference, mostly due to the unpredictable nature of ED work. Based on our context-specific findings, further research might address factors influencing work-life balance in ED physicians with a mixed-methods approach for identification of relevant associations and intervention approaches in this field.

## Introduction

Throughout the last decades, medical professionals' work-life balance became a widely discussed topic in health policy and research in countries of the global North [1, 2]. Work-life balance was broadly described as a variable psychological state of individuals, which mirrors their perceived accordance between the investment of finite personal resources and the respective achievement of important personal goals in work and non-work domains [3]. External as well as factors pertinent to the individual were determinants of the perceived work-life balance at a given point in time [3]. Regarding the work sphere, job demands, e.g., cognitive demands and time pressure, negatively influenced perceived work-life balance while job resources, e.g., job flexibility in work times and social support by colleagues and supervisors, supported more positive evaluations of work-life balance [4]. Increasing interest in this topic in clinical medicine was traced back to the growing number of female medical students and practicing physicians

anonymization of participants in full transcripts cannot be fully guaranteed due to a small group size. On this account, our ethic approval does not allow the release of full transcripts. Further, the participants themselves did not agree in their informed consent with data sharing of full transcribed texts. However, data excerpts can be accessed upon reasonable request by contacting anna.schneider@charite.de and medsoz@charite.de.

**Funding:** The author(s) received no specific funding for this work.

**Competing interests:** The authors have declared that no competing interests exist.

[5], the changing mentality of new physician generations as well as concomitant demands for fair opportunities regarding parental leave and part-time employment during medical careers [1]. In this context, the request for more family-friendly work conditions and a better work-life balance in clinical medicine were repeatedly voiced [6, 7]. Research showed that aspects of work-life balance affect career decisions of medical students and young physicians, including the choice of specialization and job satisfaction [1, 8, 9]. Studies reported heterogeneous results concerning associations between gender and the perception of work-life balance [10–13]. Thus, female physicians were still underrepresented in higher professional positions, especially amongst clinical academics, which was attributed to higher levels of role conflict between the non-work and work domain and lacking role models [14]. Furthermore, the clinical work setting emerged as a decisive factor where physicians employed in inpatient healthcare settings [10, 15, 16] and those in university medical hospitals [17] reported significantly worse work-life balance. Comparably little is known about the work-life balance of emergency department (ED) physicians, while several studies were performed to analyze this topic in other medical specialties like trauma surgery [18], radiology or gynecology [13, 17, 19, 20]. However, ED physicians are particularly subject to a stressful and demanding work environment due to shift work, largely unpredictable and high patient turnover as well as a considerable share of time-sensitive tasks [21–23]. One qualitative interview study conducted with family members of emergency medical services (EMS) workers showed negative perceptions of the compatibility of EMS work and the private life sphere [23]. However, similar studies on ED physicians' experiences are missing to date. ED-specific research is important to identify challenges influencing work-life balance and opportunities to improve the status quo in emergency medicine. Previous studies in other settings indicated that by promoting a more family-friendly clinical work environment, mental and physical health risks due to professional burnout may decrease whilst career motivation among physicians may increase [24]. In times of an ageing medical workforce and staff shortages, well-designed work systems may increase the attractiveness of the medical profession for future generations and may help to retain highly-qualified personnel [25, 26].

The lack of current, content-rich research on the everyday experiences of work-life balance in ED physicians provided the backdrop of our research. By applying an explorative approach with qualitative focus groups, we aimed to identify a wide set of ED-specific challenges and opportunities for work-life balance as well as ED physicians' suggestions for practical approaches to improve the compatibility of work and family as well as private life.

## Materials and methods

The structure of the Methods section is oriented towards the *Consolidated criteria for reporting qualitative research (COREQ)* checklist [27], i.e., characteristics of the research team and their relationship with study participants, aspects of the study design as well as the analytical approach of our study.

### Research team and reflexivity

The head physician of both study sites gave the impetus to investigate aspects of work-life balance in physicians working in EDs. After joint consultation, the conception, implementation and analysis of the study was allocated to cooperating researchers from the Institute of Medical Sociology and Rehabilitation Science. Two female employees were appointed for the realization of this focus group study, i.e., one doctorate public health researcher and a medical student in her 7th semester who worked as an undergraduate assistant in the ED. In preparation of the study, both were trained and advised by an experienced senior researcher with a

background in sociology. Both moderators demonstrated prior experience with research projects in the emergency medical setting, which induced that some of the study participants were familiar with their names and occupational roles before focus groups. However, moderators refrained from sharing their personal goals and reasons for conducting research on work-life balance in the ED so that it seems unlikely that participants held preconceptions about their roles or opinions concerning this topic.

## Study design

The target population consisted of physicians who worked at one of two ED sites of a major university hospital in a capital city during the time of data collection. We applied purposive sampling with regard to the professional position of potential participants (stratified into residents/ specialists vs. senior physicians) and with regard to parenthood (stratified into parent vs. no parent) to enable the collection of relevant and rich data with regard to our research question. We excluded physicians from other hospital wards who underwent a six- to twelve-month mandatory rotation in the ED as part of their residency. This exclusion criterion was applied to ensure sufficient knowledge of and familiarity with ED work conditions among participants. A study information sheet and informed consent were sent via email to all eligible ED physicians with the request to participate in the study. The contact information of one of the moderators was provided in the email for potential inquiries. Out of 18 physicians who were invited to take part in the study, 14 agreed to participate. Reasons for non-participation were not sampled. After physicians' consenting feedback, participants were divided into four groups, consisting of three to four physicians each. Participants' gender was collected but not used for assignment to groups.

Focus groups were carried out between May 2019 and March 2020 at both ED sites located on two different campuses of the university hospital. Focus groups were held in seminar rooms spatially in or close to the respective EDs in order to increase the accessibility for participation during or after the physicians' work shifts. Upon arrival at the seminar room, participants chose a seating position at an arranged rectangular table facing each other. The moderators were seated next to each other. Participants had the opportunity to ask questions regarding the study content and data protection measures at the beginning of each focus group. Pastries and beverages were offered. No further compensation was provided for study participation. No other persons besides the participants and the two moderators were present during focus groups. The study was approved by the ethics committee of the Charité–Universitätsmedizin Berlin (EA2/135/19) and was presented to the general works council of the hospital. Written informed consent was obtained from all participants.

The fourteen final participants were diverse regarding their professional position (n = 5 residents, n = 2 specialists, n = 7 senior physicians), their gender (n = 7 females, n = 7 males) and with regard to their characteristic of being a parent (n = 8 yes, n = 6 no). The time frame of one hour was set at the beginning of each focus group. The average duration of a focus group was 52 minutes. The total duration of all focus groups was 209 minutes. One of the following stimuli on the topic of work-life balance was presented at the start of each focus group to initiate discussion between participants:

*Stimulus 1: "(. . .) but for example Denmark (. . .) there is no way that a physician cannot go home on time to pick up his children from daycare (. . .)", "(. . .) we can talk about the compatibility [of family and work] but in everyday clinical routine it can often not be implemented, except when you would ditch the patient but, thankfully, that does not happen." What do you associate with this quote?*

*Stimulus 2*: *"As a childless young assistant, I was always suspicious of these colleagues who were parents. They never had time, always had to go home when I was still working on my 27th medical report, were always already finished with their stuff, had concepts in the twinkling of an eye, and never came along to the pub. I had no understanding of kids and even less of parents." What thoughts and own experiences do you associate with this quote*?

Stimuli depicted an interview quote from a previous research project and a blog quote. In the first focus group, the stimuli and guideline were tested and deemed suitable for application. Stimulus 1 was chosen for focus groups with specialized and senior physicians. For the focus group with residents only, stimulus 2 was applied since moderators considered it the better fit. Moderators did not interrupt participants during their conversation and kept their participation during focus groups to a minimum. If the discussion held off for multiple seconds, moderators asked questions that followed a prepared guideline (see S1 Table). The guideline focused on the following topics: (1) current experiences of participants regarding work-life balance, (2) comparison between their situation now and in prior professional positions, (3) comparison of their current with other workplaces and (4) ideas and improvement measures concerning work-life balance in emergency medicine.

## Data analysis

All focus groups were recorded with a solid-state digital tape recorder. Additionally, moderators took anonymized field notes to describe the spatial setup and the interaction between participants during focus groups. Transcription of audio tapes was performed loosely based on the rules of Dresing and Pehl [28]. All person-related data was pseudonymized in the transcription process to impede inferences about individual participants. We did not provide participants with transcripts for comment or correction after focus groups to minimize workload for participants. A word-processing program was applied for data management. The methodological orientation of this study was based in content analysis. Hence, final transcripts were analyzed using the qualitative content analysis approach described by Kuckartz [29]. To derive categories from the material, transcribed speech was separated into paragraphs with similar content. In the next step, the paragraphs were reduced to specific key lines that reflected the main statement of this section. In the following, categories were assigned to the paragraphs based on their key lines. In the beginning, the transcript of one focus group was used to create a category tree that was applied to the remaining transcripts and continuously improved during this process. Another author analyzed the transcripts of ten-minute sequences of each focus group to cross-control the coding strategy. Subsequently, discrepancies and similarities were discussed to optimize the derived codes and themes. Preliminary and final findings were repeatedly discussed between co-authors in different constellations in order to integrate multiprofessional perspectives from emergency medicine, sociology, public health and health services research. Superordinate topics used for the following presentation of results were structured based on the focus group guideline. However, we derived all themes from data only. Due to the small study group and data protection measures, results are presented across focus groups and not for individual groups. Furthermore, the topic of data saturation was discussed before study start but the small target group of ED physicians prevented researchers from conducting more than the four realized focus groups. Nevertheless, current methodological research suggests that four focus groups and an inclusion of participants from all relevant strata regarding the research question is sufficient for achieving code saturation, i.e., to capture the breadth of a topic [30], which corresponded with our exploratory aim to identify relevant topics regarding work-life-balance in ED physicians.

## Results

We categorized qualitative data into the following three superordinate categories: (1) past work-life balance, (2) present work-life balance and (3) suggestions to improve work-life balance of ED physicians in the future. All identified sub-categories of the three main categories are listed in S2 to S4 Tables with representative quotes. The main findings are contextualized below.

### Past work-life balance of emergency physicians

Participants discussed the topic of work-life balance by drawing comparisons between their respective past and current experiences in different clinical environments. The discussion particularly evolved around their past and current work in the field of emergency medicine as well as in the more general university medical context. Generally, participants described a predominantly positive temporal development of the state of work-life balance in emergency and university medicine. Physicians reported beneficial factors for a better work-life balance in the past. However, they further described numerous past work conditions and work structures, which hampered work-life balance in comparison to current ED work.

### Past work conditions and aspects of the work environment

According to participants, the past work environment was characterized by a strong, inner-clinical hierarchical structure which restricted physicians' work-life balance decisively. Relevant decisions concerning the scheduling of shifts were predominantly made by the department head and some senior physicians. Thus, participants reported a lack of possibilities to express wishes in the scheduling of their shifts in the past, which they contrasted with current practice in the ED. Participants described the past resulting lack of flexibility as a reason for diminished work-life balance, mostly for young and female physicians. Furthermore, the planned and actual duration of shifts was longer compared to today's shifts. The number of physicians scheduled per shift as well as in the core team was lower. In the past, overtime was common and taken for granted by most junior and senior physicians. Further, the stigmatization of male physicians who expressed claims for parental leave was illustrated as a problem. However, contrary to these statements, participants highlighted that a stronger solidarity and social bond among physicians facilitated work-life balance in the past. High personnel turnover in university hospitals was designated as a reason for weaker and transient social bonds between young physicians working in the ED.

### Past physician's mentality and behavior regarding work-life balance

Participants reported that physicians showed a stronger tendency to prioritize their work instead of their family or private life in the past. Individual motives as well as external expectations of the department head and senior physicians were discussed as possible reasons. In this context, a shift in priority from work towards family and private life among present young physicians was identified. Further, in the past, younger physicians pursued less claims regarding adequate work times, workload, flexible work time models or participation in work scheduling. In addition, participants underlined that a variety of employment laws were already in effect in the past. However, physicians often did not make use of them due to worries about possible negative consequences for their training or career. Consequently, participants emphasized that part-time work and longer periods of parental leave were rarely claimed. Especially during residency, a reduction of work hours with the aim to focus on childcare was unusual.

Today, however, reductions and sick days in case of acute health problems of the child, are common practice and more self-evident for both female and male young physicians.

## Past context factors of emergency medicine and university medicine

Participants reported that a high patient turnover, along with staff shortages, caused a high workload for physicians in the past. This indirectly restricted work-life balance. However, some participants claimed lower ED patient numbers in the past. Organizational support structures for physicians with children were described as minimal in the past. In contrast to today, physicians could not resort to support from so-called "hospital father's representatives" when considering parental leave or work time reduction or when addressing topics of work-life balance with their superiors. According to participants, these now available structures are of great relevance for physicians.

## Present work-life balance of emergency physicians

When discussing current conditions that influence work-life balance, participants shared positive and negative experiences from their everyday work. The maintenance of work-life balance was seen as highly dependent on available private support by partners as well as the age and stage of career of the individual physician, respectively. Work-life balance was described as being achievable for physicians with leadership roles rather than for young physicians during residency.

## Present work conditions and aspects of the work environment

Several work conditions were described to be directly linked to the compatibility of a career and family or private life. Physicians positively highlighted the current ED practice to be involved in work scheduling which allowed physicians with children to better plan ahead work and family time. Further positive factors mentioned in this context were (a) the scheduling of two days off in between the switch from night to the late or early shift, (b) the restriction of night shifts to four, or maximum five in number, and (c) the availability of an on-call service to coordinate the work schedule in case of staff shortages due to illness. According to participants, the latter practice allowed for a more balanced distribution of shifts among staff by avoiding that the same physicians fill in too often. The total increase in the department's staffing of the last few years was also positively emphasized.

On the other hand, the common occurrence of overtime hours was presented as highly incompatible with family and private life. Reasons for overtime included deficits in the handover of patients between shifts, which were described as unstructured and frequently interrupted. Reduced physician staffing of the night compared to the late shift was also reported as a common cause for overtime work in the late shift. Participants discussed that overtime not only reduced family time, but also increased the risk for negative physical and psychological health effects, e.g., insomnia and exhaustion. By hampering everyday planning, overtime resulted in challenges of organizing childcare and leisure activities. Furthermore, the aspect of missing regular work breaks was described as problematic, which was associated with a nonexistent "culture of taking breaks" among ED physicians. This observation deviated from participants' narratives of other clinical specialties with shift work systems where work breaks were coordinated and staff adhered to them more strictly. The shift work system itself was another relevant factor influencing work-life balance in participants' point of view. Reported advantages of shift work included the possibility to hand over patients to the next shift. This practice would theoretically result in fixed work hours and a good planning capability for physicians. However, shift work-pertinent holiday and weekend work reduced work-life balance. Also,

participants described that adhering to appointments regarding childcare, e.g., visits to the pediatrician or school events, was often complicated since shifts cannot be individually shortened. Generally, the remuneration of shift work was described to be inadequate when considering the high work burden and the accompanying physical and mental health impairments in affected physicians. The only possibility to get past the shift work system, and to profit from more flexibility and family-friendly work hours, was seen in the professional advancement to the position of a senior physician. Finally, another significant work demand was seen in the lacking overall hospital patient transfer system. This process obliged ED physicians to make numerous phone calls to transfer ED patients to internal and external hospital wards. Participants associated this practice with a delay in patient treatment and with unnecessary frustration, stress and conflicts in ED physicians.

Participants underlined that trusting interpersonal work relationships indirectly accounted for a family-friendly workplace. These opened up the possibility to ask colleagues for support if needed, e.g., to switch shifts or to ensure leaving work on time. The current ED work atmosphere was described as naturally tolerant towards physicians with children. However, some participants apprehended that physicians without children would fill in more often and work more often on holidays. The relationship between physicians with and without children was also described to differ with regard to the individual competitiveness and solidarity of physicians. However, for both groups, participants mentioned that perceived high expectations regarding their availability and work effort would cause pressure, which negatively affected private life. Furthermore, work-life balance was related to restrictions in the planning of clinical rotations for residents working part-time. Participants pointed out that these disadvantages caused delays in their residency progress and negatively affected future career chances.

## Present physician's mentality and behavior regarding work-life balance

Participants reflected that specific personality features were more common among physicians than other professional groups. These characteristics included a high degree of dedication, the will to sacrifice for patients and a high sense of responsibility towards colleagues. The wish of younger physicians to represent high-performing individuals in the workplace was seen as a means to leave a good impression on colleagues and to increase the prospect of an extension of a fixed-term employment contract. Causes for overtime work were thus also seen in physicians' feeling of responsibility and solidarity as well as the tendency to feel guilty when leaving work on time. Participants summarized that more experienced physicians, in contrast to their younger colleagues, knew about the operating principles of the hospital system. This knowledge would make it easier to evaluate task priority and to effectively manage work time. Both factors were considered to reduce the risk of overtime hours. Differences in the mentality of younger and older physicians were also discussed. In this context, older physicians showed the tendency of practicing and expecting more overtime hours from colleagues. This practice was believed to still set an example to the younger generation of physicians.

## Present context factors of emergency medicine and university medicine

Participants described ED-specific characteristics such as high patient turnover, unpredictable patient volumes and frequent work interruptions. These work requirements hampered time slots for organizational tasks (e.g., documentation, student teaching, and research) during regular work hours and thus resulted in overtime work. With regard to work-life balance, participants recognized that their employer offers family-friendly structures. These included childcare with long opening hours and flexible pick-up times, parent's representatives or the possibility to claim flexible worktime models. However, some participants criticized that the

actual chance to receive a place in the hospital-owned daycare center was very low. Furthermore, fixed-term employment contracts were described to be family-unfriendly as they would come along with worries and fears about the future career.

To evaluate ED structures in a greater context, participants drew comparisons with (a) other EDs, (b) with other clinical workplaces and disciplines, and (c) with jobs outside the healthcare sector. Some participants discussed similarities and differences between the two ED sites of the study hospital from their own experience. Based on that comparison, participants concluded that one ED site was characterized by less overtime hours and workload, which was attributed to the availability of a patient manager responsible for patient transfers and better handover practices. Furthermore, one ED employed more balanced shift work times, which, according to participants, caused less potential for overtime work. Based on their work experience at other wards, participants stated that structures increasing family-friendliness in other wards with shift work systems were clear work assignments, strict regulations concerning work times and breaks, and higher staffing in every shift. Participants also explained that, compared to the ED, patient numbers were foreseeable and staff fluctuations less common. Both improved work-life balance in the long run. However, contrary experiences of chaotic and unfair work scheduling at other wards were also shared. Regarding international comparisons, participants stated that inner-clinical patient transfer structures were partly better regulated in EDs of other countries. They speculated that this circumstance was based on the consensus of ED patients' high priority and their immediate entitlement to inpatient admission. Other statements concerning clinical work abroad dealt with the lower rates of overtime work in countries like Denmark, the lower prioritization of work and, generally, the higher relevance of work-life balance for employees. Accordingly, physicians with children would be granted more days off in other countries. Furthermore, participants emphasized that other countries provisioned a bigger pool of substitute physicians in case of sick leave. With regard to other professions, participants reflected that jobs outside of the healthcare sector are characterized by longer employment contracts, more structured work breaks and less workload. This was regarded as being more compatible with family and private life than in health professions.

## Improvement suggestions for work-life balance in emergency physicians

When reflecting on possible ways to address the above-mentioned challenging work conditions with the aim to increase work-life balance, participants underlined the importance of an open and honest exchange between junior and senior ED physicians. Furthermore, the strengthening of the importance of the ED in the context of university hospital structures was considered as an important step to allow for better work conditions for ED physicians.

## Suggestions for the improvement of work conditions and aspects of the work environment

To simplify work scheduling in cases of high illness rates and, simultaneously, in order to increase the attractiveness of ED work, suggestions to increase the pool of part-time workers were made. This measure was also seen as a way to reduce the workload of each individual physician. According to participants, the promoting of punctual shift ends should be emphasized, irrespective of the family, partner or child status of each physician. Overtime work should therefore become an exception rather than the rule. Means to achieve fixed work times are undisturbed, fast and punctual patient handovers between shifts, e.g., by reinforcing the intermediate shift overseeing patient care during handovers. Furthermore, an increase of staffing in the night shift was discussed as a strategy to avoid overtime hours in the late shift. Participants further described that a higher variability in everyday tasks may decrease the risk of

monotonous work. The implementation of a stricter "culture of taking breaks" by setting up break times for each shift, as far as patient volumes allow, was seen as a possibility to create recreational opportunities during high workload.

Certain privileges for physicians in the first two years after the birth of a child were discussed, including an additional day off in each annual quarter or one shorter shift per month. Acceptance and tolerance towards physicians with children were considered crucial to increase family-friendliness at the ED workplace. Especially for young physicians, demands for equal chances during residency were phrased, which would not depend on their child status or their work model. These included fair conditions regarding clinical rotations. Finally, participants expressed their wish for more appraisal interviews with the department head to foster a trusting relationship and to offer regular opportunities to discuss current problems concerning one's own work-life balance.

### Suggestions for the improvement of context factors in university medicine

Providing more places in hospital-owned daycare centers was described as an important support structure for physicians with children working in the shift system. Furthermore, general claims for more vacation days and a reduction of total monthly work time for shift workers were expressed. Likewise, an increase in remuneration and more long-lasting employment contracts were voiced. Participants also strongly advocated for the setup of a coordinated hospital-wide patient transfer structure.

Results of qualitative content analysis regarding discussed aspects of ED physicians' work-life balance in sub-categories and themes are summarized in (Fig 1).

## Discussion

Our qualitative focus group study presents rich information on factors affecting work-life balance among ED physicians and, simultaneously, suggests pathways from the physician's point of view to improve work-family fit in emergency medicine. The main discussed topics

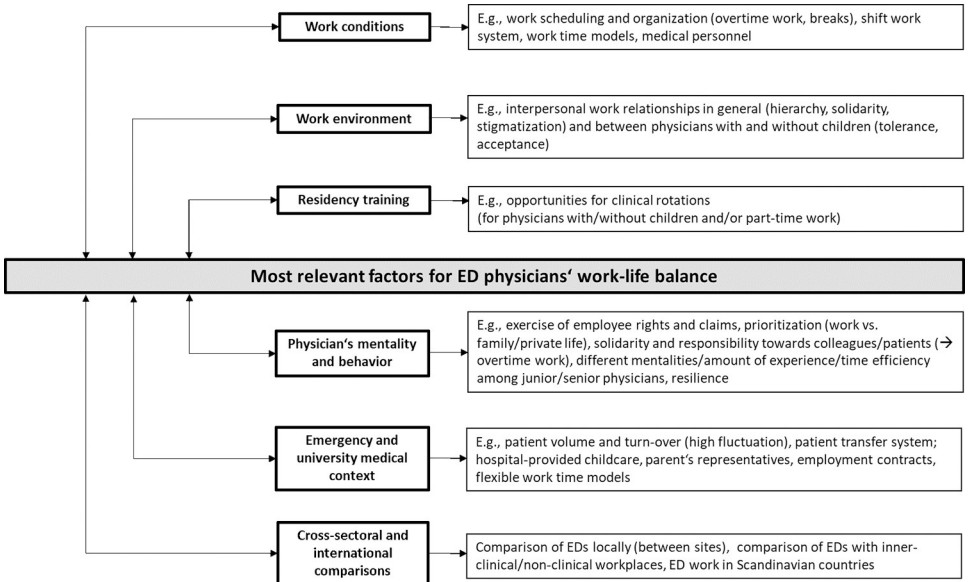

**Fig 1. Pertinent aspects of ED physicians' past and present work-life balance and avenues for future improvement.**

included ED work conditions and the work environment. Furthermore, aspects of residency training, physician's mentality and behaviors as well as context factors of emergency medicine and university medicine in general were addressed. Additionally, participants drew comparisons with other work fields. This was done to evaluate benefits and disadvantages of different practices and work structures and their effect on work-life balance. Our findings indicate that the compatibility of work and private and family life in ED physicians can be promoted by offering family-friendly organizational support structures but also by assuring beneficial work conditions. This applies to participation opportunities in duty scheduling, social support by colleagues and supervisors, and the opportunity for regular work breaks. Participants' comparisons between the prior and present state of work-life balance in emergency medicine point to a mainly positive development to date. However, current structures still fail to fully meet the existing demands and wishes of ED physicians. This is likewise reported for physicians practicing in other medical disciplines [31].

Participants saw a major challenge in balancing shift work with family and private life, which was also described as a conflict area in EMS personnel [23]. They discussed career advancement to the position of a senior ED physician as the only option to exit the ED shift work system. Still, concerns about the physical and mental health impairment resulting from permanent shift work were expressed by ED physicians. These statements are in line with existing research on the adverse effects of shift work regarding physical and mental health as well as medical staffs' intention to leave clinical practice [16]. Despite these challenges, our participants also saw positive chances in shift work, including the possibility to hand over their patients to the subsequent shift and, therefore, leave the ED on time. Still, high numbers of overtime hours and a concomitant deficit of regular work breaks were described as frequent experiences, especially by young ED physicians. Overtime and missing work breaks were associated with lower job satisfaction and work-life balance by participants. Besides the demand for standardized patient handover processes, the optimization of work scheduling was seen as a way to improve work-life balance. Solution approaches for work scheduling in EDs were recently described, emphasizing the need for special work scheduling algorithms in the ED based on indicators of high patient turnover and temporal fluctuations [32]. Changes in work schedules were also discussed as a way to increase work-family-compatibility and the general attractiveness of the field of emergency medicine for young physicians [32]. Besides the above-mentioned challenges of ED work, its advantages were also discussed in focus groups. They included the possibility to conduct a high variety of tasks on a daily basis. This aspect was described as a strength of emergency medicine in previous research [33]. Consistent with statements from participants, work scheduling should therefore assure that physicians experience this variability in their everyday work to reduce repetitiveness.

A recent meta-thematic synthesis of factors affecting clinical academic pathways reported that especially female physicians face different barriers regarding career advancement pertaining to role conflicts between work and non-work domains [14]. Respective factors were socio-cultural expectations regarding their role as primary child caregivers, lacking female role models in leadership positions in clinical medicine, and missing organizational opportunities for part-time work. Although most of these factors were not explicitly described by our participants as current problems of their work environment, the narrative of an out-of-hours culture in the department was often picked up as a central theme by participants in the form of extensive overtime hours. Furthermore, concerns over equal opportunities for part-time residents were raised. They may indicate the beginnings of the above-described barriers for (mostly female) physicians regarding career advancement at the stage of completion of medical training in university hospitals [14]. The positive effects of organizational structures enabling part-time work and the importance of the availability of flexible working models for work-life

balance in physicians were further underlined in various studies [10, 19, 34, 35]. Our participants also highlighted these options as additional ways to decrease work burden for each individual physician and decrease role conflict between work and non-work domains. Previous research focusing on female physicians reported that the availability of part-time work as well as maternal leave was seen as a way to increase the attractiveness of hospital work for this target group [36]. However, our participants pointed out the risk of experiencing disadvantages during residency in case of opting for part-time work due to family responsibilities. Consequently, the very existence of family-friendly hospital and departmental structures does not guarantee their actual utilization and acceptance [37]. A way to overcome respective disadvantages for residents during training could lie in the promotion of respective open conversations among junior and senior physicians. This wish was also expressed by our participants. Higher work-life-balance can be achieved when employees can flexibly choose their work time models to accommodate resources for the (temporal) care of children or relatives [35]. Furthermore, our results revealed that young physicians experienced the highest work-life-interference. Therefore, they require the most assistance. Similarly, career support and counselling were described as a necessity in previous studies, also among medical students with children [38].

The promotion of work environments that are premised on the acceptance and tolerance as well as equal rights for physicians with and without children was discussed as a further way to increase work-life balance. One suitable approach to increase job satisfaction and foster a trusting organizational culture might be the offer of regular appraisal interviews with the department head [31]. This point was also discussed by our participants. Furthermore, the implementation of long-term contracts was associated with a reduction in job insecurity and higher job satisfaction. Previous studies show that short-term contracts are especially present in university hospitals [31] and cause high stress levels in employees [39]. Expanding the duration of employment contracts may also be a way to decrease perceived performance pressure and overtime work. This may especially apply to young physicians, who often feel the need to stay longer as they try to avoid negative consequences for their residency or contract renewal. Generally, our participants marked that collegial relationships have suffered from the increase of medical staff fluctuation in university medicine. Considering that working in an effective core team was described as a major benefit of ED work [40], this topic needs to be addressed in the future. Amongst others, interventions including individual and team training programs could be a way forward [37]. On the organizational level, this may include measures to address cultural factors, such as the review of an out-of-hours culture, or an adequate take on individual needs, e.g., the provision of support for part-time workers. These interventions can take effect in addressing pertinent barriers at the individual organization despite the presence of larger societal factors impeding work-life balance [14].

Comparisons drawn by our participants with other work fields underlined that the employment in hospitals generally comes with a strong work-life-interference. This finding is consistent with existing literature [10, 15, 16]. Nevertheless, also in the clinical context, work-life balance was described as highly variable, i.e., depending on the discipline, the availability of personnel, the mentality and personal support structures of each individual physician and the overall organizational structures. Studies with medical staff from different disciplines support this theory [17, 20, 31]. Participants drew specific comparisons with Scandinavian countries to demonstrate that medical systems with a high work-life-compatibility are already practiced in other countries. However, a study has shown that the fear of not being able to balance work and family spheres was also present among female Swedish doctors [41]. These differences in perception support the need for international comparative research regarding work-family-fit in emergency medicine, which do not exist so far. The demand for more daycare places was a discussion point in our focus groups, too, supporting similar demands in previous research

[13, 38]. Further suggestions for improvement, such as facilitating structured patient management and transfer from the ED were proposed. Personal experiences shared by participating physicians revealed that workplaces which already realized such practices, allowed a better work-family-fit for their employees through the reduction of workload and overtime hours.

With the challenges associated with the SARS-CoV-2 pandemic, a shift towards more work-life-interference in emergency medicine seems likely [42]. Emerging factors influencing work-family-fit should thus be evaluated and adapted to present challenges. Following previous research and results from our work, interventional studies are in need to evaluate the benefits and limits of measures aimed to improve work-life balance in ED staff. For that, future quantitative research could offer additional valuable insights by numerically weighing the most relevant aspects regarding work-life balance in ED physicians.

## Limitations

Our study provides detailed insight into opinions and everyday experiences of ED physicians concerning the compatibility of their work and family and private life. With respect to its content, study format and the target group of ED physicians, it is the first study of this kind. We promoted open discussions between participants with only minimal input and guidance by moderators. Consequently, physicians' statements vary in their focus and description, which complicates a stringent summary of the main factors of work-life balance versus general challenges of the ED work environment and thus to grade topics based on their relevance for the primary research question. We refrain from statements about the frequency of specific topics due to the small number of participants and potential data protection issues and due to our explorative approach. Furthermore, by assembling focus group participants on the basis of their career stage and parenthood, we intended to facilitate an open discussion between participants without potential worries about negative consequences due to organizational hierarchies or other prejudices. However, this approach limits inferences on negotiation processes of particular topics between physicians across career stages and different family structures.

## Conclusions

Higher participation in work scheduling as well as the availability of hospital-owned childcare are two out of several factors currently contributing to increased work-life balance in ED physicians. However, a high work burden and health implications resulting from overtime hours or shift work still constitute factors of a family-unfriendly work environment. From our findings, we conclude that working in an ED still results in a comparatively high work-life-interference compared to other (non)clinical fields, mostly due to the unpredictable nature of emergency medicine. Based on these context-specific qualitative findings of an exploratory study, further research projects might address factors influencing work-life balance in ED physicians with a mixed-methods approach including qualitative methods to deepen the understanding of discussed relevant topics as well as quantitative methods, e.g., employee surveys, in order to quantify the most prominent key factors and associations for interventional measures.

## Supporting information

**S1 Table. Focus group guideline questions.**
(DOCX)

**S2 Table. Synthesized results of discussions on "Past work-life balance of emergency physicians" in sub-categories and themes.**
(DOCX)

**S3 Table. Synthesized results of discussions on "Present work-life balance of emergency physicians" in sub-categories and themes.**
(DOCX)

**S4 Table. Synthesized results of discussions on "Improvement suggestions for work-life balance in emergency physicians" in sub-categories and themes.**
(DOCX)

**S1 File. Consolidated criteria for reporting qualitative research (COREQ).**
(DOCX)

## Acknowledgments

The authors would like to thank all participating physicians for their time and interest in the study.

## Author Contributions

**Conceptualization:** Samipa Pudasaini, Liane Schenk, Martin Möckel, Anna Schneider.

**Data curation:** Samipa Pudasaini.

**Formal analysis:** Samipa Pudasaini, Anna Schneider.

**Investigation:** Samipa Pudasaini, Anna Schneider.

**Methodology:** Samipa Pudasaini, Liane Schenk, Anna Schneider.

**Project administration:** Samipa Pudasaini, Anna Schneider.

**Supervision:** Liane Schenk, Martin Möckel.

**Validation:** Samipa Pudasaini, Liane Schenk, Martin Möckel, Anna Schneider.

**Visualization:** Samipa Pudasaini, Liane Schenk, Martin Möckel, Anna Schneider.

**Writing – original draft:** Samipa Pudasaini.

**Writing – review & editing:** Samipa Pudasaini, Liane Schenk, Martin Möckel, Anna Schneider.

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
