## [Decision Letter · Decision Letter 0]

26 Jul 2022

PONE-D-21-38549Work-life balance in physicians working in two emergency departments of a university hospital: Results of a qualitative focus group studyPLOS ONE

Dear Dr. Schneider

Thank you for submitting your manuscript to PLOS ONE. After careful consideration, we feel that it has merit but does not fully meet PLOS ONE’s publication criteria as it currently stands. Therefore, we invite you to submit a revised version of the manuscript that addresses the points raised during the review process.

We look forward to receiving your revised manuscript.

Kind regards,

Soham Bandyopadhyay

Academic Editor

PLOS ONE

Journal Requirements:

Reviewers' comments:

Reviewer's Responses to Questions

**Comments to the Author**

1. Is the manuscript technically sound, and do the data support the conclusions?

Reviewer #1: Partly

2. Has the statistical analysis been performed appropriately and rigorously? 

Reviewer #1: N/A

3. Have the authors made all data underlying the findings in their manuscript fully available?

Reviewer #1: No

4. Is the manuscript presented in an intelligible fashion and written in standard English?

Reviewer #1: Yes

5. Review Comments to the Author

Reviewer #1: This study comprehensively highlights the various facets of work life balance among ED physicians. A few suggestions to improve this paper are given.

In the methodology, it is not necessary to mention as first author etc. Simply mentioning that a medical student, a Post grad trainee is enough.

In line 175, it is not necessary to mention which author did the process. Instead mention that transcribed speeches were divided according to similar content and so on.

It would be good if a paragraph containing the demography of each focus group was mentioned along with whether discussion was initiated with scenario 1 or 2.

The results are very descriptive and lengthy but reflect valuable points. It would be good if some measure (proportion or percentage) could be included to salient points highlighting how many of the focus groups mentioned a particular factor. This would demarcate more important widely mentioned points from points which were mentioned by one or two people.

Perhaps sentences could be shortened with more paragraph breaks to facilitate better understanding (making discussion and results a bit more concise)

6. PLOS authors have the option to publish the peer review history of their article (what does this mean?). If published, this will include your full peer review and any attached files.

Reviewer #1: No

---

## [Author Response · Author response to Decision Letter 0]

5 Sep 2022

Dear Dr Soham Bandyopadhyay, Dear reviewers,

Thank you very much for the opportunity to submit a revised version of our manuscript entitled “Work-life balance in physicians working in two emergency departments of a university hospital: Results of a qualitative focus group study”. 

Foremost, we would like to express our sincerest gratitude for the thoughtful reviewer comments and helpful recommendations. We carefully revised the manuscript with respect to all the points raised by the reviewer. Please find below a point-by-point statement of the changes we made to the manuscript. Our responses are inserted in italics. Changes to the previous version are highlighted in red font in the marked-up copy of the manuscript. 

Regarding your remark about data availability, we would like to point out that our focus group transcripts contain person-related sensitive information. Due to the low number of participants, a possible identification cannot be fully ruled out, also after pseudonymization of data. Moreover, since one study objective was to reflect critically on current working conditions, the prospect of publishing the interview recordings would have significantly limited willingness to participate. Further, the participants did not agree with sharing fully transcribed texts in their informed consent and our ethic approval does not allow the release of all transcripts. However, data excerpts can be shared upon reasonable request by contacting anna.schneider@charite.de and medsoz@charite.de as described in our revised data availability statement.

We sincerely hope that we have adequately addressed all of the issues raised and that the revised manuscript now meets your as well as the reviewer’s approval. We are looking forward to your feedback.

With best regards, 

Samipa Pudasaini, Liane Schenk, Martin Möckel, Anna Schneider

Journal Requirements:

Authors: We have applied all style requirements based on the PLOS ONE template and hope to now meet the journal’s demands on this matter.

Authors: Thank you for this note. As described in our response letter, legal as well as ethical restrictions both apply to our study and prohibit the publishing of full transcripts. First, the participants did not agree to that in their informed consent. Also, the ethical approval only allows us to share parts of the transcribed material since a full anonymization of the participants cannot be guaranteed due to the small group size. Based on this, we would be grateful if you could update our data availability statement to the following: 

“Data cannot be shared publicly because of legal and ethical restrictions. Our focus groups contained person-related sensitive information and the anonymization of participants in full transcripts cannot be fully guaranteed due to a small group size. On this account, our ethic approval does not allow the release of full transcripts. Further, the participants themselves did not agree in their informed consent with data sharing of full transcribed texts. However, data excerpts can be accessed upon reasonable request by contacting anna.schneider@charite.de and medsoz@charite.de.”

Authors: We thank you for this remark. In point two, we have explained in detail the legal and ethical restrictions that prohibit the publishment of full transcripts. Certainly, researchers can send data sharing requests to anna.schneider@charite.de and medsoz@charite.de if interested.

Authors: We thank you very much for this offer. We would be grateful if you could update our data availability statement as described in point two.

Reviewer’s comments:

1. Is the manuscript technically sound, and do the data support the conclusions?

Reviewer #1: Partly

Authors: We have carefully revised the detailed comments of the reviewer and hope that the conclusions we have drawn are now more traceable and coherent.

2. Has the statistical analysis been performed appropriately and rigorously? 

Reviewer #1: N/A

3. Have the authors made all data underlying the findings in their manuscript fully available?

Reviewer #1: No

Authors: The data used includes potentially sensitive information and person-related data and based on the participants’ informed consent and the ethical approval, does not allow full publishment. In our revised data availability statement (see point two), we have now described this matter in detail and hope for your understanding.

4. Is the manuscript presented in an intelligible fashion and written in Standard English?

Reviewer #1: Yes

Authors: We thank you very much for this positive evaluation. 

5. Review Comments to the Author

Reviewer #1: 

5.1. This study comprehensively highlights the various facets of work life balance among ED physicians. A few suggestions to improve this paper are given.

Authors: We thank you for the positive response and the given suggestions for improvement. Based on these remarks, we have revised our manuscript and hope to now fully meet the criteria for publication.

5.2. In the methodology, it is not necessary to mention as first author etc. Simply mentioning that a medical student, a Post grad trainee is enough.

Authors: We agree with the reviewer and have corrected the respective sections in the methodology by removing this information (see lines 91-94).

5.3. In line 175, it is not necessary to mention which author did the process. Instead mention that transcribed speeches were divided according to similar content and so on.

Authors: As suggested, we have changed this sentence by removing the information about who transcribed the material (see lines 172-178).

5.4. It would be good if a paragraph containing the demography of each focus group was mentioned along with whether discussion was initiated with scenario 1 or 2.

Authors: We fully understand the relevance of including this information. Therefore, in the method section, we have now inserted an additional paragraph with a general description of which stimulus was used for groups with specialized or senior doctors versus assistant doctors (see lines 152-154). More precise quantitative information on the professional position, gender and child status of each individual focus group can unfortunately not be provided since statements listed in Supplementary tables 1-3 are marked with the focus group number (FG 1-4) they were obtained from. Based on that and due to the low number of participants, a retracing of statements to specific doctors could not be ruled out when listing the detailed demography of all focus groups separately. We hope for your understanding in this matter.

5.5. The results are very descriptive and lengthy but reflect valuable points. It would be good if some measure (proportion or percentage) could be included to salient points highlighting how many of the focus groups mentioned a particular factor. This would demarcate more important widely mentioned points from points which were mentioned by one or two people.

Authors: We thank you for this remark. Our work focuses on depicting valuable statements and experiences shared by the participating doctors by choosing a qualitative approach. Unfortunately, neither the nature of the sample construction nor the sample size allow for quantifying statements. We share the curiosity about the statistical relevance of the statements, but this must be reserved for future research. Methodologically, we assume to reconstruct typical orientations that are not represented by only one individual. Further, weighing the statement in a quantitative manner does not follow the standardized protocol of explorative qualitative research, as e.g. described by Tenny et al. (1). In our discussion, additionally to our past statement regarding the quantification of results (see lines 556-559), we have now embedded an additional sentence highlighting the major importance of analyzing this topic in a quantitative manner in future research (see lines 545-547).

5.6. Perhaps sentences could be shortened with more paragraph breaks to facilitate better understanding (making discussion and results a bit more concise).

Authors: As marked in the revised manuscript, we have shortened sentences throughout the results and discussion section. With that, we hope to have significantly improved the readability of our work.

References cited in the response to reviewers

1. Tenny S, Brannan GD, Brannan JM, Sharts-Hopko NC. Qualitative Study. StatPearls. Treasure Island (FL)2022.

---

## [Decision Letter · Decision Letter 1]

31 Oct 2022

Work-life balance in physicians working in two emergency departments of a university hospital: Results of a qualitative focus group study

PONE-D-21-38549R1

Dear Dr. Schneider,

We’re pleased to inform you that your manuscript has been judged scientifically suitable for publication and will be formally accepted for publication once it meets all outstanding technical requirements.

Kind regards,

Soham Bandyopadhyay

Academic Editor

PLOS ONE

---

## [Editor Report · Acceptance letter]

4 Nov 2022

PONE-D-21-38549R1 

Work-life balance in physicians working in two emergency departments of a university hospital: Results of a qualitative focus group study 

Dear Dr. Schneider:

I'm pleased to inform you that your manuscript has been deemed suitable for publication in PLOS ONE. Congratulations! Your manuscript is now with our production department. 

Kind regards, 

on behalf of

Dr. Soham Bandyopadhyay 

Academic Editor

PLOS ONE